# Macrophage-Derived Factors with the Potential to Contribute to Pathogenicity of HIV-1 and HIV-2: Role of CCL-2/MCP-1

**DOI:** 10.3390/v15112160

**Published:** 2023-10-27

**Authors:** Chunling Gao, Weiming Ouyang, Joseph Kutza, Tobias A. Grimm, Karen Fields, Carla S. R. Lankford, Franziska Schwartzkopff, Mark Paciga, Tzanko Stantchev, Linda Tiffany, Klaus Strebel, Kathleen A. Clouse

**Affiliations:** 1Division of Biotechnology Review and Research 1, Office of Biotechnology Products, Center for Drug Evaluation and Research, U. S. Food and Drug Administration, 10903 New Hampshire Avenue, Silver Spring, MD 20993, USA; chun.gao@fda.hhs.gov (C.G.); joseph.kutza@lilly.com (J.K.); tobiasagrimm@yahoo.de (T.A.G.); carla.lankford@fda.hhs.gov (C.S.R.L.); fschwart@web.de (F.S.); drmarkpacigaphd@gmail.com (M.P.); tzanko.stantchev@fda.hhs.gov (T.S.); linda.tiffany@fda.hhs.gov (L.T.); 2Division of Biotechnology Review and Research 2, Office of Biotechnology Products, Center for Drug Evaluation and Research, U. S. Food and Drug Administration, 10903 New Hampshire Avenue, Silver Spring, MD 20993, USA; 3Laboratory of Molecular Microbiology, National Institute of Allergy and Infectious Diseases, NIH, Bethesda, MD 20892, USA; kstrebel@nih.gov

**Keywords:** HIV-1, HIV-2, CCL2, STAT1, macrophage, CULLIN 2, RBX1

## Abstract

Human immunodeficiency virus type 2 (HIV-2) is known to be less pathogenic than HIV-1. However, the mechanism(s) underlying the decreased HIV-2 pathogenicity is not fully understood. Herein, we report that β-chemokine CCL2 expression was increased in HIV-1-infected human monocyte-derived macrophages (MDM) but decreased in HIV-2-infected MDM when compared to uninfected MDM. Inhibition of CCL2 expression following HIV-2 infection occurred at both protein and mRNA levels. By microarray analysis, quantitative PCR, and Western blotting, we identified that Signal Transducer and Activator of Transcription 1 (STAT1), a critical transcription factor for inducing CCL2 gene expression, was also reduced in HIV-2-infected MDM. Blockade of STAT1 in HIV-infected MDM using a STAT1 inhibitor significantly reduced the production of CCL2. In contrast, transduction of STAT1-expressing pseudo-retrovirus restored CCL2 production in HIV-2-infected MDM. These findings support the concept that CCL2 inhibition in HIV-2-infected MDM is meditated by reduction of STAT1. Furthermore, we showed that STAT1 reduction in HIV-2-infected MDM was regulated by the CUL2/RBX1 ubiquitin E3 ligase complex-dependent proteasome pathway. Knockdown of CUL2 or RBX1 restored the expression of STAT1 and CCL2 in HIV-2-infected MDM. Taken together, our findings suggest that differential regulation of the STAT1—CCL2 axis may be one of the mechanisms underlying the different pathogenicity observed for HIV-1 and HIV-2.

## 1. Introduction

Human immunodeficiency virus types 1 (HIV-1) and 2 (HIV-2) are lentiviruses known to infect humans [1,2]. HIV-1 and HIV-2 share many similarities, including their basic gene arrangement, modes of transmission, intracellular replication pathways, and clinical consequences. Both infections may progress to acquired immunodeficiency syndrome (AIDS), which has been known to be one of the diseases with high levels of mortality and morbidity throughout the world for several decades [3]. However, HIV-2 infection is more geographically restricted and is less pathogenic compared to HIV-1 infection [4,5]. Significant differences exist between the two infections in multiple components of the immune system. The immune response to HIV-2 appears more protective against disease progression, suggesting that pivotal immune factors limit viral pathogenesis, most likely as a result of decreased virulence and/or better control by the host immune system. However, the factors responsible for the HIV-2 decreased pathogenicity are still inadequately clarified [6,7]. If such immune responses could be replicated or induced in HIV-1-infected patients, they might extend survival and reduce requirements for antiretroviral therapy. Studying the decreased pathogenicity of HIV-2 is not only important to provide insight regarding this disease, but it is likely to provide insight for the treatment of HIV-1 and other lentivirus diseases.

Infection with HIV generally starts with mild symptoms, followed by a slow progression towards advanced HIV disease and immunosuppression. HIV-infected macrophages serve as major reservoirs for the virus in tissues of the human body [8,9]. The innate immune system senses HIV infection and can initiate antiviral responses by multiple nucleic acid sensors, including toll-like receptors (TLRs) 3, 7, 8, and 9, interferon (IFN)-inducible protein (IIP) 16, and cyclic GMP-AMP synthase [10]. Recognition of HIV by these pattern recognition receptors leads to induction of antiviral and proinflammatory responses via activation of NF-κB and interferon regulatory factors (IRFs) signaling pathways, ultimately resulting in induction of the production of cytokines, chemokines, and type I IFNs [11]. Type I IFNs act as the first line of defense against viral infections by mediating immunoregulatory, growth-inhibitory, and antiviral activities through activation of the JAK/STAT signaling pathway and the induction of a multitude of interferon-stimulated genes (ISGs) that encode antiviral proteins [11,12]. Chemokines are shown to have dichotomous effects on HIV replication [13]. On the one hand, ligands for CCR5 (RANTES, MIP-1α, and MIP-1β) and CXC chemokine receptor 4 (CXCR4) (stromal-derived factor-1; SDF-1) have been demonstrated to inhibit HIV entry into CD4^+^ T cells and PBMCs, as well as monocytic and CD4^+^ T cell lines. On the other hand, β-chemokines may facilitate viral reservoir establishment by recruiting macrophages to the infection site and promoting HIV replication in monocytes and macrophages. Among the β-chemokines, CCL2 (MCP-1) has shown the most robust enhancing effect on HIV-1 viral replication and pathogenesis [7,14,15]. HIV-1 infection upregulates expression of both CCL2 and its receptor CCR2, and high levels of CCL2/CCR2 are observed in HIV-1-infected patients.

In this study, we found that CCL2 expression is enhanced in HIV-1-infected MDM but is suppressed in HIV-2-infected MDM. The inhibition of CCL2 expression in HIV-2-infected cells correlates with a reduction in STAT1 via the CUL2/Rbx1-dependent proteasome pathway. These findings suggest that the STAT1—CCL2 axis is differentially regulated by HIV-1 and HIV-2 infection and may play a role in controlling HIV-1 and HIV-2 pathogenesis.

## 2. Materials and Methods

### 2.1. Monocyte Isolation and Differentiation

Leukapheresis blood units from healthy seronegative donors were obtained from the Department of Transfusion Medicine (DTM) under an Institutional Review Board (IRB)-approved protocol at the National Institutes of Health. A categorical exemption is in place for experimental studies performed by CDER/FDA researchers using existing deidentified blood donors who provided written informed consent according to the international ethical guidelines for biomedical research involving human subjects. These samples were anonymized prior to being sent to the FDA. Peripheral blood mononuclear cells (PBMC) were subsequently isolated from the leukapheresis unit by density gradient centrifugation. Monocytes were purified by countercurrent centrifugal cell elutriation as previously described [16]. Over 90% of the cells in the elutriated monocyte fraction were CD14^+^, as determined by Flow Cytometry analysis, and showed ≥95% viability (trypan blue exclusion test). Monocytes were differentiated in culture for 5–7 days at 37 °C in 5% CO_2_ at a concentration of 2 × 10^6^ cells/mL, 2.0 mL per well, in 6-well tissue culture plates (Costar, Cambridge, MA, USA) using DMEM (Invitrogen Corporation, Carlsbad, CA, USA) complete medium containing 10% pooled human serum (PHS), 2 mM L-glutamine, 1 mM sodium pyruvate, and penicillin (50 units/mL)/streptomycin (50 µg/mL) (Invitrogen) to generate human monocyte-derived macrophages (MDM). 

### 2.2. HIV-Infection of Human Monocyte-Derived Macrophages

MDM were harvested by scraping and then plated into 24-well tissue culture plates (Nunc, Inc., Naperville, IL, USA) at a concentration of 0.5 × 10^6^ cells/mL, 1.5 mL per well. After 24 to 48 h of culture, MDM were infected with HIV-1 isolates or primary isolates of HIV-2 that had been minimally passaged in human PBMC [17]. HIV-1_BaL_ (Catalog# 10-177-000) was purchased from Advanced Biotechnologies (Columbia, MD, USA). HIV-1_ADA_ (Catalog# 10-255-000) was obtained through the AIDS Research and Reference Reagent Program (ARRRP), NIAID, NIH, from Dr. H. Gendelman [18,19], then expanded in human macrophages prior to purification by ultracentrifugation and cryopreservation (Advanced Biotechnologies) as described previously [16]. HIV-1_UG24_ (Catalog# 1649) and HIV-1_BCF03_ (Catalog# 3335) are minimally passaged HIV-1 isolates that were obtained from the AIDS Research and Reference Reagent Repository. HIV-2_B2-B5_ and HIV-2_B7-B9_ isolates were from Portugal and described previously [17], and the stocks of these HIV-2 isolates were expanded in primary human PBMC as previously indicated [16]. Infection with HIV-1 was performed as previously described [16,19]. For infection with HIV-2, input virus for the primary HIV-2 isolates was equalized based on reverse transcriptase (RT) activity using 0.1 cpm/cell. Virus adsorption was carried out for 4 h at 37 °C, and cells were then washed, and fresh medium was added. Every 3 days after infection, 80% of the culture medium was collected, stored at −80 °C, and replaced with fresh medium.

### 2.3. Reverse Transcriptase (RT) Assay

The progression of viral replication in MDM infected with HIV-1 or HIV-2 was monitored by measuring RT activity using a ^3^H-based method described by Hoffman, et al. [20]. The RT value shown is the average of duplicate samples (cpm/25 µL) that differed by not more than 15%.

### 2.4. Cytokine ELISA Assays

ELISA kits from R&D Systems (Minneapolis, MN, USA) were used to quantitate the levels of CCL2 (Catalog# DCP00), CCL3 (Catalog# DMA00), CCL4 (Catalog# DMB00) and CCL5 (Catalog# DRA00B) produced throughout the time course of the infections following the manufacturer’s instructions.

### 2.5. Relative Quantitative Real-Time PCR

Total RNA was extracted from cells using TRIzol reagent (Invitrogen; Waltham, MA, USA), and 2 μg RNA was reverse transcribed to cDNA using Omniscript (Qiagen; Germantown, MD, USA) and a random hexamer primer method. *CCL2* primers and probe were designed by primer express 3.0 software (Applied Biosystems; Foster City, CA, USA), of which the forward primer sequence was 5′- GAA GAA TCA CCA GCA GCA AGT G-3′, the reverse primer was 5′-GAT CTC CTT GGC CAC AAT GG-3′, and the probe was 6′-FAM-5′-AAG AAG CTG TGA TCT TCA A-3′. The probe of *CCL2* spans the border of exons 2 and 3 to exclude DNA contamination. The *STAT1* primer and probe mixture were obtained from Invitrogen (Catalog# Hs01013998_ml). Each 30 μL of PCR reaction included 2 μL of 2× diluted cDNA (equal to 100 ng RNA), 300 nM sense and antisense primers, 200 nM 6′-Fam labeled fluorescent target probe, and 1.5 μL 20× glyceraldehyde-3-phosphate dehydrogenase (GAPDH) control probe. Amplification of the endogenous control and target gene was performed in the same well to standardize the amount of the expression target RNA. The CT (cycle threshold) data, defined as the number of PCR cycles required for the fluorescent signal to cross the threshold as a positive reaction, was used to determine ΔCT and ΔΔCT for the relative amount of the target mRNA. The formula 2^−ΔΔCT^ was used for calculation of CCL2 and STAT1 gene expression. The ΔCT is obtained by subtracting the GAPDH CT value from the target gene CT value. The ΔΔCT was obtained by subtracting the control ΔCT from the target ΔCT (Relative quantitative PCR calculation, Invitrogen).

### 2.6. Western Blot Analysis of Protein Expression 

Total protein was extracted with complete lysis-M buffer from Roche Life Science (Indianapolis, IN, USA) that included a proteinase inhibitor mixture. The total target protein (20 µg) was separated on a 4–12% NuPAGE gel to detect STAT1 using a rabbit anti-STAT1 antibody purchased from Cell Signaling Technology (Danvers, MA, USA) (Catalog# 9172) as a probe. Actin expression was measured using a mouse anti-actin antibody from Sigma (St. Louis, MO, USA) (Catalog# A2228) and used as a sample loading control.

### 2.7. Construction of STAT1-Encoding Pseudotyped Retrovirus and Transduction of MDM

The entire human *STAT1-* encoding region was amplified by RT-PCR using the forward and backward primer pair: 5′-AGATCTGTCTCAGTGGTACGAACTTCAG-3′ (Forward) and 5″-GTTAACCTATACTGTGTTCATCATACTG-3′ (backward). The STAT1 cDNA fragment was cloned into the Qiagen PCR cloning vector, confirmed by DNA sequencing, and then inserted into the retroviral vector, pMIG-w. The final construct was designated as pMIG-STAT1, which was co-transfected with a Gag/Pol packaging plasmid and VSV-G into HEK293T cells by lipofectamine 2000 (Invitrogen) to produce pseudo retrovirus. The expression of STAT1 in HEK293T cells was confirmed by Western blotting (Appendix A). The STAT1-encoding pseudotyped retrovirus was used to transduce MDM that were infected with or without HIV-1 or HIV-2 isolates by the spin transduction method.

### 2.8. Gene Knockdown by Small Interfering RNAs

Small interfering RNAs (siRNA) specifically targeting human *RBX1* (ID# s19386), *CUL1* (ID# 139193), *CUL2* (ID# 139190), *CUL3* (ID# 139187), *CUL4a* (ID# 139184), *CUL4b* (ID# 13299), *CUL5* (ID# 139086) and *CUL7* (ID# 14847) were purchased from Invitrogen. The siRNAs were transfected into uninfected and HIV-2-infected MDM using lipofectamine RNAiMAX reagent (Thermo Fisher, Waltham, MA, USA) following the manufacturer’s instructions. 

### 2.9. Statistical Analysis

Statistical analysis was performed using the Student’s *t*-test method, and the statistical significance of any difference was determined by a *p*-value < 0.05.

## 3. Results

### 3.1. HIV-2 Infection Suppresses CCL2 Production in Human MDM

HIV-1 infection of human MDM has been reported to induce the production of β-chemokines [21]. To investigate the potential correlation between β-chemokine induction and the reduced pathogenicity observed with HIV-2, we first performed an initial study to analyze β-chemokine expression, measuring levels of the chemokines CCL2, CCL3, CCL4, and CCL5 in supernatants harvested from HIV-1- and HIV-2-infected MDM cultures at the peak of viral replication. In the pilot study, we infected human MDM with seven previously described, minimally passaged HIV-2 isolates: HIV-2_B2-B5_ and HIV-2_B7-B9_ [17], as well as two CCR5-tropic lab adapted and 2 minimally passaged (one CXCR4-tropic and one CCR5-tropic) HIV-1 isolates, and then compared the profile of chemokines released by the infected MDM (Table 1). Although the levels of chemokines induced by HIV-1 or HIV-2 showed some variabilities in MDM derived from different healthy donors, as would be expected, specific profiles of chemokine production emerged depending on the HIV-1 or HIV-2 isolates (Table 1, Figure 1). Moreover, we noticed that, when compared to the constitutive production of CCL2 observed with uninfected MDM, MDM infected with HIV-2 isolates produced reduced levels of CCL2, but MDM infected with HIV-1 isolates produced levels of CCL2 that were increased above those observed for uninfected control MDM (Table 1). In contrast, there were generally no notable differences in the levels of CCL3, CCL4, or CCL5 produced by uninfected, HIV-1- or HIV-2-infected MDM (Table 1), although a slight increase in CCL3 and CCL4 was observed for the donor shown in Table 1 when MDM were infected with lab-adapted HIV-1_Bal_. The levels of chemokines produced by HIV-1- and HIV-2-infected MDM did not directly correlate with the levels of viral replication (Table 1). The differential chemokine induction profiles also failed to correlate with the co-receptor involved in HIV entry, given that HIV-1_ADA_, HIV-2_B3, B4_, and HIV-2_B7-B9_ all use CCR5 for entry and clearly differ in their induced chemokine production profile (Table 1).

To confirm the differential regulation of CCL2 expression by HIV-1- and HIV-2-infected MDM, we performed a time-course study in which supernatants were harvested at different time points following infection with the seven HIV-2 and four HIV-1 isolates for assessment of the levels of CCL2 and viral replication. As shown in Figure 1A–D, active replication of HIV-1 in MDM from donor #8 augmented CCL2 expression, while active replication of HIV-2 suppressed CCL2 expression. Of note, replication of both HIV-1 and HIV-2 exhibited isolate-to-isolate and donor-to-donor variations. The CXCR4-tropic HIV-1 isolate UG24 replicated poorly in MDM from donor #8 but robustly in MDM from donor #7 (Figure 1B, Appendix A). Replication of HIV-1 UG24 and other HIV-1 isolates correlated with CCL2 production in the infected MDM (Figure 1A, Appendix A), suggesting that robust replication may be required for the HIV-1 isolates to induce CCL2 production in MDM. HIV-2 isolates B2, B8 and B9 also showed variable replication in MDM from donors #5, #6, and #8 (Figure 1D, Appendix A). In contrast to HIV-1 isolates, replication of HIV-2 isolates did not show a strong correlation with the changes in CCL2 expression levels of the infected MDM. Minimal replication of HIV-2 B2, B8, and B9 isolates in MDM from donor #8 and low replication of HIV-2 B8 and B9 isolates in MDM from donor #5 resulted in substantial suppression of CCL2 production (Figure 1C, Appendix A). 

Given that limited stocks of macrophage-tropic HIV isolates were available, we used representative isolates in the follow-up confirmatory and mechanistic studies. Statistical analyses of the data from eight healthy donors indicated that the levels of CCL2 production released by HIV-1-infected MDM were significantly higher than levels of CCL2 production constitutively expressed by uninfected control MDM, as well as levels of CCL2 released by HIV-2-infected MDM (Figure 1E). Consistent with the results showing different levels of CCL2 protein detected in the supernatants, human MDM infected with HIV-1 and HIV-2 isolates also expressed different amounts of CCL2 mRNA (Figure 1F), suggesting that regulation of CCL2 expression by HIV infection occurs at the gene transcription level.

### 3.2. STAT1 Reduction Is Associated with Inhibition of CCL2 Production in HIV-2-Infected MDM 

We next investigated the potential molecular mechanism for CCL2 inhibition in HIV-2-infected MDM. Given that the regulation of CCL2 occurs at the gene transcription level, we reasoned that HIV-2 infection may inhibit signaling pathways that play a critical role in activating CCL2 gene transcription. To identify the signaling pathways that are differentially impacted by infection with HIV-1 and HIV-2, we compared the gene expression profiles of HIV-1- and HIV-2-infected MDM by Affymetrix microarray using cell samples prepared at the peak of viral replication. Among the differentially expressed genes, we were particularly interested in the STAT1 gene that was identified by Affymetrix microarray to be upregulated in HIV-1-infected MDM but downregulated in HIV-2-infected MDM (Figure 2A). STAT1 has been reported to bind to the *CCL2* locus and function as a key regulator of CCL2 gene expression in the human monocytic THP-1 cell line [22,23]. By RT-PCR and Western blotting, we confirmed the opposite effects of HIV-1 and HIV-2 acute infection on the expression of STAT1 in MDM. Both STAT1 mRNA and protein levels were increased above the constitutively expressed levels observed in uninfected media controls for HIV-1-infected MDM but were reduced relative to uninfected controls in HIV-2-infected MDM (Figure 2B,C). 

To further confirm the role of STAT1 in regulating CCL2 expression in human MDM infected with HIV-1 and HIV-2, we performed loss-of-function and gain-of-function studies. In the loss-of-function study, infected MDM were cultured with the STAT1 inhibitor, fludarabine, to block STAT1 function. In the gain-of-function study, infected MDM were transduced with pseudo-retrovirus expressing human STAT1. Culture of the MDM with fludarabine for 6 h significantly reduced the production of CCL2 in both HIV-1- and HIV-2-infected MDM (Figure 3A). In contrast, the transduction of STAT1-expressing, but not the control, pseudo retrovirus increased the production of CCL2 in HIV-2-infected MDM (Figure 3B). Taken together, these results suggest that STAT1 is associated with the activation of CCL2 gene expression in HIV-infected MDM.

### 3.3. STAT1 Reduction in HIV-2-Infected MDM Is Regulated by the Cullin2/RBX1 Proteasome Pathway

STAT1 has been reported to be degraded by the ubiquitination—proteasome pathway, in which viral components such as paramyxoviral V protein and RSV NS protein can function as a component of the Cullin (CUL)-RBX1 ubiquitin E3 ligase (CRLs) complex, playing an essential role in the process of STAT1 polyubiquitination and degradation by the proteasome [24,25]. In addition, HIV Vif, Vpr, Vpu, and Vpx can also act as a component of the CRLs complex [26,27,28,29,30]. The HIV Vpx/CUL4/DCAF1/RBX1 complex relieves inhibition of HIV-1 infection of macrophages by mediating SAMHD1 protein degradation [27,28]. Based on these previous findings, we hypothesized that STAT1 may also be degraded by the ubiquitination—proteasome pathway, which depends on a viral component from HIV-2 but not HIV-1. To test this hypothesis, we transfected HIV-2-infected MDM at the peak of viral replication with control siRNA or siRNA specifically targeting human CUL 1, 2, 3, 4, 5, 7, or RBX1 and harvested cell culture supernatants at 72- and 96-h post-siRNA transfection to assess the expression levels of CCL2 by ELISA. Statistical analysis of results from 5–11 donors indicated that HIV-2 infection reduced the levels of CCL2 in the cells transfected with control siRNA, and knockdown of CUL2 or Rbx1 significantly upregulated the expression of CCL2 in the infected cells at both 72- and 96-h time points, which correlated with the restoration of STAT1 protein levels in the cells (Figure 4 and Appendix A). These results support our hypothesis that STAT1 may be degraded by the CUL2-Rbx1 E3—proteasome pathway, which requires a viral component(s) existing in HIV-2 but not present (or not active) in HIV-1.

## 4. Discussion

Although infection of humans with either HIV-1 or HIV-2 can ultimately cause AIDS and exhibit similar clinical features, infection with HIV-2 is geographically restricted, occurs less frequently, and is generally less pathogenic than HIV-1 [4,31]. The progression to AIDS in HIV-2-infected individuals is also reduced and/or delayed, with a corresponding mortality rate about two-thirds lower than that observed in HIV-1-infected individuals [2,31,32,33]. Our group is particularly interested in understanding the regulation of HIV pathogenesis and the mechanism(s) underlying the reduced pathogenicity of HIV-2. 

Recognition of the HIV virus by pattern recognition receptors of innate immune cells induces antiviral and proinflammatory responses via activation of JAK/STAT signaling, NF-κB, and interferon regulatory factors (IRFs) pathways, subsequently leading to the production of cytokines and chemokines that facilitate or suppress HIV virus infection and/or replication [10,12]. We hypothesized that infection with HIV-1 and HIV-2 may induce distinct expression profiles of cytokines and/or chemokines, which could influence pathogenesis and determine the course and outcome of the disease. 

HIV-1 infection of human macrophages leads to the release of β-chemokines including CCL2, CCL3, CCL4, and CCL5 [34,35,36]. Although HIV-2 has been reported to infect human macrophages [37,38], the effects of this virus on chemokine production by macrophages have not been fully investigated. In this study, we found that CCL3, CCL4, and CCL5 β-chemokines are induced similarly in MDM infected with HIV-1 and HIV-2. In contrast, CCL2 expression that is constitutively expressed at high levels in non-infected, cultured macrophages is further enhanced subsequent to HIV-1 infection of MDM but is generally suppressed in HIV-2-infected MDM. The CCL2/CCR2 axis plays a critical role in the pathogenesis of HIV-1 infection [14]. Elevated serum levels of CCL2 show a positive correlation with in vivo HIV-1 viral load [21] and disease progression, as well as the development of HIV-associated dementia [39]. Although initially considered as a chemoattractant primarily for monocytes, CCL2 was subsequently found to also attract activated and memory T lymphocytes [40,41]. CCL2, along with other members of this family, has been shown to increase the replication of X4-tropic strains of HIV-1 in activated CD4^+^ T lymphocytes [42]. More recently, it was determined that CCL2 has the capacity to increase X4-tropic HIV-1 entry into resting CD4^+^ T cells [43]. This means that CCL2 produced by HIV-1-infected macrophages would have the capacity to recruit both monocytes and activated T cells to the site of infection, facilitate the latent infection of resting T cells, and enhance virus replication in activated T cells. Thus, the shutdown of CCL2 production at the transcriptional level would create a great deterrent to the infection of both macrophages and T cells by limiting the recruitment of susceptible cells. It would also impair the establishment of virus reservoirs involving cells from both lineages because monocyte recruitment would be diminished, and the entry of X4 tropic virus into resting CD4^+^ cells would be reduced. Therefore, inhibition of CCL2 expression in HIV-2-infected macrophages may serve as one of the mechanisms underlying the reduced pathogenicity of HIV-2.

Although the levels of chemokines produced by HIV-1- and HIV-2-infected human MDM did not always correlate with the levels of viral replication (Table 1), this type of variability, as well as variabilities among and between donors, has been previously reported [44,45]. In MDM from some donors, minimal or low replication of HIV-2 isolates B2, B8, and B9 was able to markedly suppress CCL2 production. The underlying mechanism warrants future investigation. In addition, the differential chemokine induction profiles also failed to correlate with the specific co-receptor involved in HIV entry. 

Gene expression profiling of HIV-1- and HIV-2-infected MDM revealed that STAT1 is differentially expressed between MDM infected with HIV-1 and HIV-2. STAT1 has been reported to activate CCL2 gene expression in the human THP-1 monocytic cell line. By performing both loss-of-function and gain-of-function studies, we demonstrated that STAT1 also plays a role in regulating CCL2 expression in HIV-infected MDM. 

STAT1 is a member of the STAT transcription factor family that acts both as signal transducers in the cytoplasm and as activators of transcription in the nucleus [46]. STAT1 is the major mediator of the cellular response to IFNs. Upon IFN stimulation, STAT1 is activated by tyrosine phosphorylation within the cytoplasm by a class of non-receptor tyrosine kinases called Janus kinases (JAKs and TYK2) associated with IFN receptors. Phosphorylated STAT1 molecules form dimers through reciprocal phosphotyrosine (pTyr)-SH2 interactions and are translocated to the nucleus, where they activate gene transcription. Given that HIV infection induces the release of IFN, which activates STAT1, and STAT1 positively regulates the gene expression of itself [47], a potential explanation for the reduction of STAT1 in HIV-2-infected MDM is that HIV-2 is less capable of inducing IFN than HIV-1. Our preliminary study indicates that MDM may not express dramatically different levels of IFNs following acute infection with HIV-1 and HIV-2. However, this should be confirmed by further studies. 

In addition to gene regulation, STAT1 protein can also be degraded by the polyubiquitination—proteasome pathway [24,25,48,49]. Reduction in STAT1 protein level subsequently impacts the expression of its regulated genes, including itself. The mechanism of STAT1 reduction, mainly occurring at the protein level, may explain the difference between the changes of CCL2 and STAT1 mRNA levels in HIV-2-infected MDM (Figure 1F and Figure 2B). Polyubiquitination is an enzymatic protein modification mediated by ubiquitin-activating enzymes (E1), ubiquitin-conjugating enzymes (E2), and ubiquitin ligases (E3). The Cullin (CUL)–Rbx E3 ubiquitin ligases (CRLs) are the largest superfamily of E3 ubiquitin ligases, with more than 400 members in mammals [50]. Mammals express seven canonical Cullin proteins (Cul1, Cul2, Cul3, Cul4A, Cul4B, Cul5, and Cul7) that form multi-subunit CRLs, designated as CRL1–7. Each CRL has four core components: a Cullin protein that acts as a scaffold to the CRL, a RING finger protein (Rbx1 or Rbx2) that binds to an E2 ubiquitin-conjugating enzyme, a substrate receptor that recognizes the target protein via a specific domain called degron, and adaptor proteins that bridge the substrate receptor to the Cullin [50]. CRLs are often hijacked by viruses to evade the immune system and promote replication by mimicking the substrate receptors [11,51]. STAT1, as a key factor in controlling viral infection, can be degraded by CRLs, in which Paramyxoviruses V protein and respiratory syncytial virus nonstructural proteins serve as substrate receptors [25,49]. HIV Vif and Vpx are also reported to act as a component of CRLs [26,27,28]. In this study, we identified that CUL2 and RBX1 are involved in the negative regulation of STAT1 in HIV-2-infected MDM. Interestingly, CUL1 knockdown also restored CCL2 in HIV-2-infected MDM from some of the donors, even though the overall restoration of CCL2 is not statistically significant following analysis of the data from 6 donors. A definite conclusion of CUL1 involvement in regulating CCL2 in HIV-2-infected MDM requires further study with MDM from additional donors. In addition, one critical question that remains after this study is what viral component existing in HIV-2, but not HIV-1, isolates functions as part of the CUL2/RBX1 E3 ligase complex. Another critical question that warrants further investigation is whether HIV-2 loses its ability to inhibit CCL2 production later in the disease process, thereby contributing to the progression to AIDS. 

In conclusion, we uncovered a STAT1—CCL2 axis that is specifically suppressed in human macrophages following HIV-2 infection. Given the role of CCL2/CCR2 and STAT1 in regulating the pathogenesis of HIV [7,14,15,52,53], these findings provide a potential explanation for the reduced pathogenicity of HIV-2, corroborating the hypothesis that the CCL2/CCR2 axis may serve as a novel cellular target for HIV therapy [14].

## Figures and Tables

**Figure 1 viruses-15-02160-f001:**
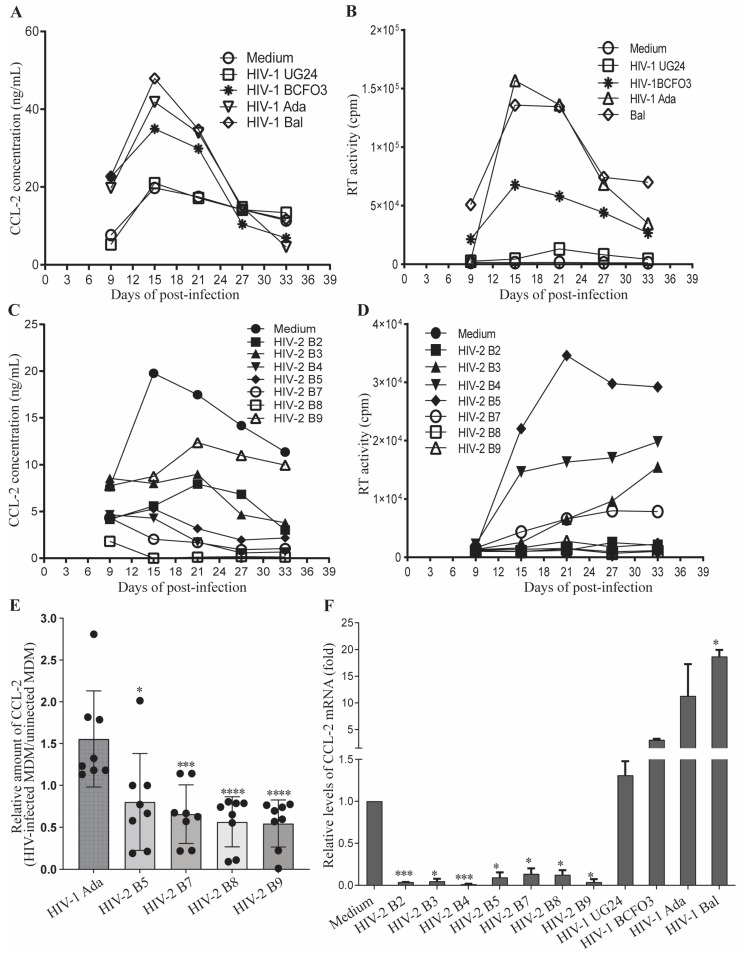
**Opposite effects of HIV-1 and HIV-2 infection on the expression of CCL2 by primary human MDM.** (**A**–**D**) Differentiated primary human MDM from donor #8 were uninfected (Medium) or infected with a variety of HIV-1 and HIV-2 strains as indicated. The production of CCL2/MCP-1 following infection with HIV-1 (**A**) or HIV-2 (**C**) in primary human MDM was determined by ELISA at the indicated time points. Data shown are representative of results from 8 healthy donors. Replication of HIV-1 (**B**) and HIV-2 (**D**) was monitored by measuring RT activity in the harvested supernatants in duplicate. RT values differed by not more than 15%. (**E**) Distinct CCL2 induction following the infection with HIV-1 and HIV-2. The concentrations of CCL2 in the supernatants harvested from MDM infected with HIV-1 Ada, HIV-2 B5, B7, B8, or B9 strains at the peak viral replication time point were determined by ELISA and normalized to the expression levels of CCL2 in the supernatants harvested from uninfected MDM. Each symbol represents an individual donor. Data were shown as the mean ± SEM (*n* = 8), and the asterisks depict significant differences between HIV-1 and HIV-2 infection groups (*n* = 8), which were analyzed using the Student’s T-test. Asterisks *, ***, and **** depict *p* ≤ 0.05, *p* ≤ 0.005 and *p* ≤ 0.001, respectively. (**F**) HIV-2 infection inhibited CCL2 gene transcription. The expression levels of CCL2 mRNA were determined by quantitative RT-PCR. The abundance of the mRNA was calculated using the comparative CT method 2^−ΔΔCT^, and the expression levels of the CCL2 gene were normalized to the expression levels of *GAPDH*. The fold changes in CCL2 mRNA expression levels for HIV-1- and HIV-2-infected primary human MDM were calculated/normalized relative to the mRNA levels in uninfected cells (Medium). Each experiment was carried out in duplicate and performed twice. Asterisks depict significant differences between uninfected (Medium) and HIV-2-infected groups; asterisks *, *** depict *p* ≤ 0.05 and *p* ≤ 0.005, respectively.

**Figure 2 viruses-15-02160-f002:**
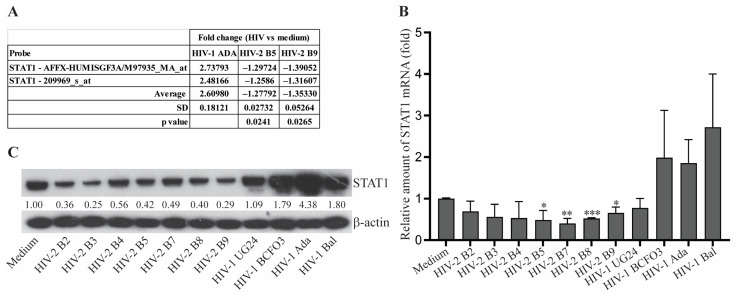
**HIV-2 infection reduces the levels of STAT1 expression in primary human MDM.** (**A**) Affymetrix microarray analysis of STAT1 gene expression in MDM infected with HIV-1 or HIV-2. Data shown represent the fold change of the microarray signal from HIV-infected MDM vs. the signal from uninfected MDM using two STAT-1 probes. (**B**,**C**) The levels of STAT1 mRNA (**B**) and protein (**C**) in uninfected (Medium) and HIV-1- or HIV-2-infected human primary MDM were assessed by quantitative RT-PCR and Western blotting, respectively. The expression levels of CCL2 mRNA were examined using cells harvested on day 15 post-infection, normalized by the levels of GAPDH mRNA, and presented as the amount relative to uninfected cells (**B**). STAT1 protein expression levels were also analyzed using cells harvested on day 15 post-infection, and the relative expression levels of STAT1 were quantified and normalized by the levels of β-actin (**C**). Asterisks *, **, and *** depict a statistically significant difference in the STAT1 mRNA levels between the untreated (Medium) MDM and HIV-2-infected MDM with *p* ≤ 0.05, *p* ≤ 0.01 and *p* ≤ 0.005, respectively.

**Figure 3 viruses-15-02160-f003:**
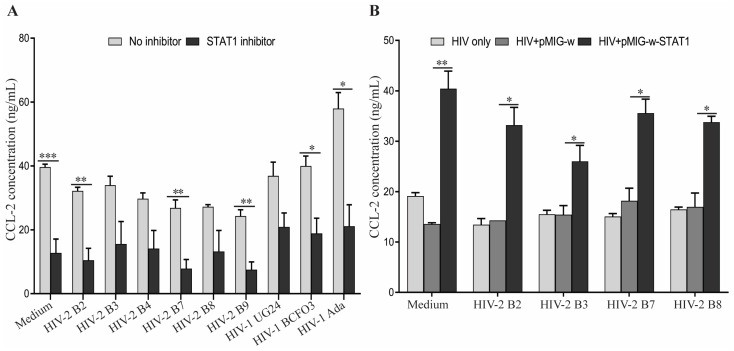
**Reduced levels of CCL2 observed following HIV-2 infection correlate with inhibition of STAT1 expression in primary human MDM.** (**A**) STAT1 blockade reduces CCL2 expression in HIV-1- and HIV-2-infected MDM. Cell culture supernatants were harvested from uninfected (Medium), HIV-1- and HIV-2-infected MDM following treatment with 0.5 µg/mL of the STAT1 inhibitor, fludarabine, for 6 h and assessed for the expression levels of CCL2 by ELISA. Asterisks depict significant differences between groups treated with or without fludarabine. (**B**) Transduction of STAT1-expressing pseudo retrovirus increases CCL2 secretion by HIV-2-infected MDM. Uninfected (Medium) and HIV-2-infected MDM were transduced with pseudo retrovirus expressing STAT1 (pMIG-w-STAT1) or the empty vector-derived pseudo retrovirus (pMIG-w). CCL2 levels were determined 48 h post-transduction by ELISA. Asterisks *, **, and *** depict a statistically significant difference in the levels of CCL2 between the compared groups with *p* ≤ 0.05, *p* ≤ 0.01 and *p* ≤ 0.005, respectively.

**Figure 4 viruses-15-02160-f004:**
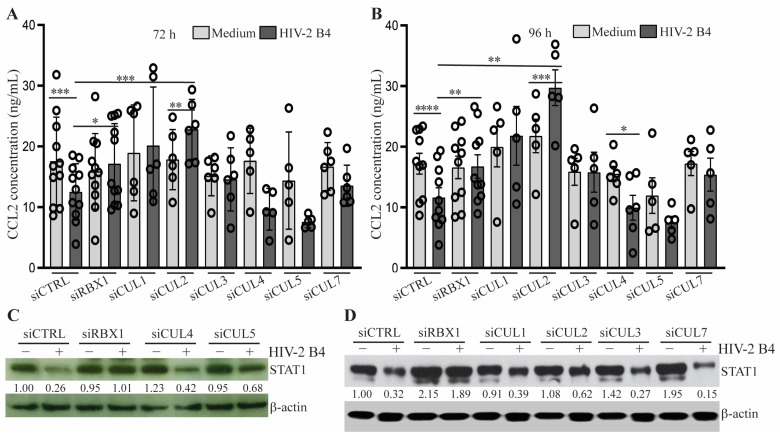
**Knockdown of CUL2/RBX1 restores STAT1 and CCL2 expression in HIV-2-infected MDM.** (**A**–**D**) Uninfected (Medium) and HIV-2-infected MDM were transfected with control siRNA or siRNA specifically targeting human RBX1, CUL1, CUL2, CUL3, CUL4, CUL5 or CUL7. CCL2 levels in the cell culture supernatants were determined 72 (**A**) and 96 (**B**) hours post-transfection by ELISA. Each symbol represents an individual donor. Data were shown as the mean ± SEM of the results from 5 donors (siCUL4 and siCUL5), 6 donors (siCUL1, siCUL3, and siCUL7), or 11 donors (siCTRL and siRBX1). Asterisks depict significant differences between the compared groups as indicated, which were determined by Student’s t-test analysis. The expression levels of STAT1 protein in uninfected and HIV-2-infected MDM following siRNA transfection for 96 h were determined by Western blotting and quantified by normalization to the levels of β-actin used as a loading control (**C**,**D**). Asterisks *, **, ***, and **** depict a statistically significant difference in the levels of CCL2 between the compared groups with *p* ≤ 0.05, *p* ≤ 0.01, *p* ≤ 0.005 and *p* ≤ 0.001, respectively.

**Table 1 viruses-15-02160-t001:** Virus replication and expression of β-chemokines in MDM infected with HIV-1 and HIV-2 isolates.

Virus	Peak RT (cpm × 10^−3^)	CCL2 (MCP-1) (ng/mL)	CCL3 (ng/mL)	CCL4 (ng/mL)	CCL5 (ng/mL)
Medium	1501.5	19.6	<0.5	<0.5	<0.5
HIV-2_B2_	9814.5	13.0	<0.5	<0.5	<0.5
HIV-2_B3_	16,117.4	8.5	0.55	0.27	<0.5
HIV-2_B4_	20,450.6	4.6	1.8	0.14	<0.5
HIV-2_B5_	42,830.1	4.2	1.4	<0.5	<0.5
HIV-2_B7_	10,966.2	4.3	1.48	0.64	<0.5
HIV-2_B8_	13,239.5	1.8	0.2	<0.5	<0.5
HIV-2_B9_	4683.8	10.7	<0.5	<0.5	<0.5
HIV-1_UG24_	11,099.8	27	0.53	<0.5	<0.5
HIV1_BCFO3_	67,695.3	34.9	0.79	0.34	<0.5
HIV-1_Ada_	136,262.7	35	1.98	0.45	<0.5
HIV-1_Bal_	144,347.2	48	5.76	3.4	<0.5

RT activity was measured every 3 days throughout the course of infection for 42 days. The peak RT value is shown in the table for each virus. The β-chemokine values were determined from the supernatants harvested at the peak of infection.

## Data Availability

The data presented in this study are available in this article as main Figures or Table.

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
