# Peer review of "Macrophage-Derived Factors with the Potential to Contribute to Pathogenicity of HIV-1 and HIV-2: Role of CCL-2/MCP-1"

_viruses, 2023, doi:10.3390/v15112160_

Round 1

Reviewer 1 Report

In the manuscript titled: “Macrophage-derived factors with the potential to contribute to pathogenicity of HIV-1 and HIV-2: Part II - Role of CCL2/MCP-1” Gao et al. continue the theme of working to determine differences that are triggered after HIV-1 infection that could differ between HIV-1 and HIV-2 to explain the relatively less pathogenic infection of HIV-2. While this work suggests that HIV-1 infection upregulates CCL-2 expression and HIV-2 expression downregulates it, it’s somewhat confusing to read because all of the isolates are not tested in all experiments, and some infect much better than others. Some of the assays show dramatic effects with isolates that appear to infect poorly.

It would be important in this work to have a control for whether the virus actually needs to infect and replicate to impact CCL-2 production and to determine the number of infected cells. Specifically, HIV-1 UG24 (Figure 1) has a very low RT signal and doesn’t impact CCL-2 levels, a plausible correlation. For HIV-2, on the other hand, B2, B8 and B9 all produced very low RT levels but all appeared to reduce CCL-2 protein and to a greater degree CCL-2 mRNA. If virus infection is causing these changes, a high proportion of the cells must be infected to produce such a dramatic change in a background of potentially uninfected cells. Otherwise there must be a high number of cells that are producing low amounts of RT or a soluble factor either with the virus or produced in response to the virus. The preparation of the viral stocks was not described. There is a reference listed for the HIV-2 isolates but is not available (missing from the publisher’s website).

STAT1 mRNA levels for HIV-2 B2, B3 and B4 don’t change significantly, yet the CCL-2 mRNA levels decreased dramatically. Is this small difference enough?

Figure 3B: The introduction of STAT1 expression vector produced a robust increase here. The pMIG-w vector offers GFP to gauge transduction efficiency. It would also be useful to see that STAT1 is being over-expressed. The decreases in CCL-2 levels are very small, so it’s not clear that STAT1 expression is overcoming much of a deficit.

Figure 4 : siRNA function should be confirmed by western blotting.

Author Response

Please see our point-to-point responses to your comments in the attached cover letter. Thank you!

Reviewer 2 Report

This manuscript by Gao and colleagues investigated the role of CCL-2/MCP-1 in HIV-1 and HIV-2 infection of MDM.  CCL2 expression was increased by HIV-1 infection, whereas it appeared to be decreased by HIV-2 infection.  Mechanistic experiments implicated STAT1 as a transcription factor involved in regulation of CCL2 expression.  Interestingly, HIV-2 infection appeared to induce proteasome degradation of STAT1, thereby reducing expression of CCL2.

This manuscript is Part II of this group’s analyses of HIV-2 infection of MDMs.  The data in general support the Authors’ conclusions.  The findings of the study are of some interest.  The merging of both Part I and Part II manuscripts into a single manuscript would make the studies more impactful and be a kindness to readers.

General Concern.

As with the Part I manuscript, the viral isolates  utilized in the study should be more carefully described, optimally in the Methods section.

Author Response

Please see our point-to-point responses to your comments in the attached cover letter (Referee #2). Thank you!

Round 2

Reviewer 1 Report

The revisions address the prior critiques adequately.